# Toxicity, Pharmacokinetics, and Gut Microbiome of Oral Administration of Sesterterpene MHO7 Derived from a Marine Fungus

**DOI:** 10.3390/md17120667

**Published:** 2019-11-26

**Authors:** Wei Tian, Liang Yang, Di Wu, Zixin Deng, Kui Hong

**Affiliations:** Key Laboratory of Combinatorial Biosynthesis and Drug Discovery, Ministry of Education, and Wuhan University School of Pharmaceutical Sciences, Wuhan University, Wuhan 430071, China; twwtss@163.com (W.T.); liangy@whu.edu.cn (L.Y.); diwu0301@163.com (D.W.); zxdeng@whu.edu.cn (Z.D.)

**Keywords:** MHO7, Log P/D values, Toxicity, Pharmacokinetics, Gut microbiome

## Abstract

Sesterterpene MHO7 derived from mangrove fungus is a novel estrogen receptor degrader for the treatment of breast cancer. To explore its safety and pharmacokinetics in vivo, Log P/D values, stability in simulated gastric/intestinal (SGF/SIF), toxicity, and pharmacokinetics studies were carried mainly by liquid chromatography technique coupled with tandem mass spectrometry (LC–MS/MS) method in mice, and the effect of MHO7 on mice gut microbiota at different time points was revealed by 16S rRNA sequencing. Log P/D values ranged 0.93–2.48, and the compound in SGF and SIF is stable under the concentration of 5 mM·L^−1^. The maximum tolerance dose (MTD) of oral administration in mice was 2400 mg·kg^−1^. The main pharmacokinetics parameters were as following: C_max_ of 1.38 μg·mL^−1^, T_max_ of 8 h, a half-life (t_1/2_) of 6.97 h, an apparent volume of mean residual time (MRT) of 8.76 h, and an area under the curve (AUC) of 10.50 h·μg·mL^−1^. MHO7 displayed a wide tissue distribution in mice, with most of the compound in liver (3.01 ± 1.53 μg·g^−1^) at 1 h, then in fat (5.20 ± 3.47 μg·g^−1^) at 4 h, and followed by reproductive organs with the concentrations of 23.90 ± 11.33 μg·g^−1^,13.69 ± 10.29 μg·g^−1^, 1.46 ± 1.23 μg·g^−1^, and 0.36 ± 0.46 μg·g^−1^ at 8, 12, 20 and 30 h, respectively. The most influenced genera of gut microbiome belonged to phylum Firmicutes (21 of 28), among which 18 genera originated from the order Clostridiales, class Clostridia, and families of Ruminococcaceae (11 of 18) and Lachnospiraceae (4 of 18). These results provide that MHO7 is suitable for oral administration in the treatment of breast cancer with the target organs of reproductive organs and regulation on Ruminococcaceae and Lachnospiraceae.

## 1. Introduction

Of the more than 80,000 identified terpenoids [1], sesterterpenoids are amongst the rarest and are an attractive source of leading compounds with little more than 1000 compounds known [2]. In recent years, amounts of biological sesterterpenoids were isolated from the metabolites of marine microorganism [3], such as neomangicols and mangicols from marine-derived fungus *Fusarium* sp. and ophiobolins from marine-derived fungus *Aspergillus* spp. [4,5,6]. Ophiobolins (Ophs) are a very rare group of sesterterpenoids that bear a unique 5-8-5 tricarbocyclic in structure [5]. At present, 72 Ophs have been reported including 35 Ophs which were produced by marine-derived fungus [5,7,8,9]. The family shows significant inhibitory effects on the growth of a variety of cancer cells, drug-resistant cells, and cancer stem cells [5]. In total, 41 cell lines have been tested for 26 Ophs with the IC_50_ values ranging from 0.08 μM to 85 ± 12 μM [5], which is considered to be potential drug candidates for cancer therapy.

In our previous study, MHO7 (6-epi-ophiobolin G, the structure is shown in Figure 1), which was produced by a mangrove fungus *Aspergillus ustus* 094102 [10], demonstrated potent antitumor activity against breast cancer cells (MCF-7) by the mechanism of down regulating estrogen receptor alpha (ERα) acting as a novel estrogen receptor degrader different from the known ERα inhibitors [11].

Although the family of Ophs have shown anticancer activity in some cancer cell lines [5], no toxicity and pharmacokinetics studies have been reported on these compounds. Due to the convenient administration, the oral route of drug administration is the most common approach comparing with injections (intravenous, intramuscular, and subcutaneous) or inhalation administration [12,13]. Drugs dissolve in the gastrointestinal tract after oral administration and then distribute throughout the body, which depends on the solubility in different pH environments of the compounds [13]. The liposolubility, safety, and pharmacokinetics study of compounds are very important in the early stage in drug development [14]. Recent studies discovered that gut microbiota influences host physiology including nutrient metabolism, resistance to infection, and immune system development [15,16]. It is also associated with diseases ranging from localized gastroenterologic disorders to neurologic, respiratory, metabolic, hepatic, and cardiovascular illnesses [15,16,17,18].

In this study, we first determined the oil/water partition coefficient in different pH environments and the stability in simulated gastric fluid (SGF) and simulated intestinal fluid (SIF) of MHO7. Then, we investigated acute toxicity of MHO7 in mice by oral administration. Based on these results, the pharmacokinetics and tissue distribution study of MHO7 in mice were carried out. Moreover, the effect of oral administration of MHO7 on gut microbiota at different time points was studied.

## 2. Results and Discussion

### 2.1. Log P and Log D Value of MHO7

Log P value of MHO7 was carried out by measuring the solubility of MHO7 in n-octanol and water system at 25 and 37 °C, respectively. The Log D values of MHO7 at physiological conditions of stomach, intestine, and plasma were examined under three pH values of 1.5, 5.0, and 7.4, respectively. The result of HPLC methodological validation is shown in Appendix A and the Log P and Log D values were calculated, as listed in Appendix A. The Log *p* values were 1.29 ± 0.05 and 2.48 ± 0.03 at 25 °C and 37 °C, respectively. The Log D values were 1.12 ± 0.03, 0.93 ± 0.02, and 1.26 ± 0.01, respectively. Proudfoot et al. (2005) reviewed the oil–water distribution coefficients of 1791 oral medicines on the market in the past 60 years and there were 37% oral medicines with the Log *p* values of 1–3 and 22.6% of 3–5 [19], illustrating that the oil–water distribution coefficient parameter of MHO7 satisfies the oral preparation.

### 2.2. Stability of MHO7 in Simulated Gastric Fluid (SGF) and Simulated Intestinal Fluid (SIF)

Simulated Gastric Fluid (SGF) and Simulated Intestinal Fluid (SIF) were prepared in the in vitro incubation system mainly containing pepsin and trypsin, respectively. Concentrations of MHO7 from each time point were obtained by liquid chromatography technique coupled with tandem mass spectrometry (LC–MS/MS) and the stability of MHO7 in SGF and SIF is shown in Figure 2, while the result of methodological validation is shown in Appendix A and Appendix A. Figure 2A exhibits that the deterioration of MHO7 reached the maximum at 2 h after addition and slowed after 2 h incubating in SGF and SIF with the concentrations of 0.05 and 0.5 mM·L^−1^. At 8 h, the percentage concentrations were 76.19 ± 0.99% in SGF and 94.83 ± 1.14% in SIF after adding MHO7 at 0.05 mM·L^−1^; however, when the initial addition concentration was 0.5 mM·L^−1^, the percentage concentrations of MHO7 were 80.2 ± 3.15% in SGF and 89.74 ± 0.86% in SIF, which revealed that MHO7 is more stable in SIF than in SGF. Figure 2B displays the effect of different concentrations of MHO7 on pepsin in SGF. The protein was precipitated immediately after adding MHO7 at 5 mM·L^−1^, which was also observed in SIF of trypsin precipitation.

### 2.3. Maximum Tolerable Dose of Oral Administration of MHO7 in Mice 

In total, 20 female and 20 male mice were used in the sighting study and main study on assessing the maximum tolerable dose (MTD) of MHO7. The MTD of MHO7 in both male and female KM mice was determined to be 2400 mg·kg^−1^ by oral administration. Within 14 days after oral administration of MHO7, no animals died, and their weight increased with time (Appendix A). There was no significant difference between the MHO7 group and control group after seven days. The weight ranges are recorded in Appendix A, which shows there was no significant difference between treatment group and control group. No obvious toxicity was observed, but a small number of animals with diarrhea was found during the first day in both female and male mice, and returned to normal within two days. The oral administration of MHO7 was well-tolerated by animals.

### 2.4. Pharmacokinetics and Tissue Distribution after Oral Administration of MHO7

A HPLC-MS/MS method for the analysis of MHO7 in mice biological samples was developed. Calibration curves of MHO7 in plasma and different tissues were constructed by plotting the peak area ratio (*Y*) of MHO7 to IS versus the nominal concentration (*X*) of MHO7 with the standard curves Y = 0.0486392 + 1.8329 × X, Correlation coefficient R^2^ = 0.9982 for plasma and Y = 0.0608319 + 1.39753 × X, R^2^ = 0.9976 for tissues. The limits of quantification of MHO7 in mice plasma and tissues were 0.5 and 2.0 nM·L^−1^, respectively. Then, the relative standard deviations (RSDs) were measured to be in the range of 4.23–6.94% for inter-day precision and 2.02–4.87% for intra-day precision. Moreover, the accuracy ranged 93.57–100.26% for inter-day and 92.50–101.15% for intra-day. In addition, the matrix effects and extraction recoveries of MHO7 in plasma and tissues were 90.05–100.63% and 90.09–93.97%, respectively. Finally, the stability under a variety of storage and handling conditions of MHO7 was stable during the routine analysis for the pharmacokinetics and tissue distribution study. Other results of methodological validation are shown in Appendix A and Appendix A.

The pharmacokinetics and tissue distribution following single oral administration of MHO7 at doses of 500 mg·kg^−1^ are shown in Figure 3. After oral administration, MHO7 was absorbed rapidly at 4–8 h and reached its peak at about 8 h with the peak plasma concentration (Cmax) 1.38 μg·mL^−1^, while the elimination of MHO7 was slow with a low concentration after 16 h (Figure 3A). The pharmacokinetics curve of MHO7 showed a little second peak at 20 h, which illustrated that the secondary absorption of MHO7 possibly occurred in plasma. Pharmacokinetic parameters achieved after oral MHO7 are shown in Table 1. The area under the plasma concentration–time curve (AUC) was 10.50 h·μg·mL^−1^ and the volume of distribution (V/F) was 479.02 L·kg^−1^. The elimination half-life (t_1/2_) was 6.97 h, indicating that MHO7 was absorbed and cleared slowly from the mice plasma. The clearance (CL/F) and mean residence time (MRT) were 476.08 L·h^−1^·kg^−1^ and 8.76 h, respectively.

The percentage of MHO7 in the stomach and intestine at different time points after oral administration of MHO7 is illustrated in Figure 3B. The percentage of MHO7 in the stomach was significantly greater than that in the intestine and reached maximum at 8 h, which was consistent with the plasma concentration–time curve. Furthermore, the amount of MHO7 in the stomach and its contents was only 16.79% of the intake, and less than 5% in the intestine and its contents. As we known, oral drugs are absorbed through the gastrointestinal tract and enter the blood circulation. Although there was a little amount of MHO7 in the gastrointestinal tract and its contents, the C_max_ of MHO7 in plasma was still low. The potential mechanism of low bioavailability might not be associated with the absorption in the gastrointestinal tract but plasma, such as a high plasma protein combination rate.

The MHO7 concentrations in other tissues were determined at 1, 4, 8, 12, 20 and 30 h after oral administration at a dose of 500 mg·kg^−1^, as shown in Figure 3C. MHO7 was widely distributed into most tissues and declined progressively with time thereafter at the concentrations of 0.36–23.90 for the reproductive organs, 0.18–11.15 for fat, 0.07–8.16 for the kidney, 0.01–3.01 for the liver, 0.08–2.34 for the lung, 0.02–1.42 for the muscle, 0.01–0.95 for the brain, 0.04–0.85 for the heart, and 0.01–0.48 for the spleen, from high to low in the order of highest concentration, which indicates that there was no long-term accumulation of MHO7 and congruent with the observed change trend in the plasma concentration. Ratios of tissues to plasma concentration at each time point are exhibited in Figure 3D. MHO7 showed wide transportability into tissues with tissue/plasma concentration ratios of > 1000 for the stomach and > 100 for the intestines, 6.05–30.45 for the reproductive organs, 0.18–17.40 for the liver, 0.68–17.24 for the kidney, 2.33–14.03 for fat, 0.28–8.55 for the spleen, 0.72–6.84 for the lung, 0.36–6.11 for the brain, 0.99–5.54 for the muscle, and 0.31–4.20 for the heart, from high to low in the order of highest radios. The corresponding concentrations values of MHO7 in tissue biodistribution and ratios of tissues/plasma each time points are summarized in Appendix A.

The pie chart of concentrations of MHO7 in tissues is exhibited in Figure 3E. At 1 h after oral administration of MHO7, highest concentration levels were observed in liver (3.01 ± 1.53 μg·g^−1^), followed by brain (0.95 ± 0.80 μg·g^−1^) and reproductive organs (0.88 ± 0.68 μg·g^−1^). At 4 h, the MHO7 was concentrated in fat (5.20 ± 3.47 μg·g^−1^), reproductive organs (2.29 ± 1.80 μg·g^−1^) and liver (1.64 ± 1.09 μg·g^−1^) in sequence. At 8 h, the highest level of MHO7 was detected in reproductive organs (23.90 ± 11.33 μg·g^−1^), followed by fat (11.15 ± 8.87 μg·g^−1^) and kidney (1.66 ± 0.74 μg·g^−1^). However, at 12 h, the levels of MHO7 from high to low were reproductive organs (13.69 ± 10.29 μg·g^−1^), kidney (8.16 ± 6.23 μg·g^−1^), and lung (2.34 ± 1.91 μg·g^−1^). Furthermore, the highest level was exhibited in reproductive organs with the concentration of 1.46 ± 1.23 μg·g^−1^ and 0.36 ± 0.46 μg·g^−1^ at 20 h and 30 h, respectively.

The results indicate that MHO7 was widely distributed in most tissues and decreased obviously after 20 h. In our previous study, MHO7 exhibited antitumor activity against breast cancer cells (MCF-7) by the mechanism of down regulating ERα [11]. Breast cancer occurs in female animals; in the pharmacokinetics experiments, especially tissue distribution, we studied whether MHO7 has potential for the treatment of breast cancer in female mice. In these organs, the concentration of MHO7 in reproductive organs was significantly higher than in other tissues during 8–30 h, which suggested that reproductive organs might be a main target organ for MHO7 with a sustained effect.

Furthermore, the concentration of MHO7 in liver at 1 h was obviously higher than in other tissues, illustrating that MHO7 metabolite in liver occurred firstly. In addition, the concentrations of MHO7 in kidney at 12 h were markedly higher than in other tissues except reproductive organs, which demonstrated the accumulation of MHO7 in kidney and suggested that renal excretion might be a main elimination route for MHO7. Finally, MHO7 exhibited a low concentration in the brain, spleen, and heart, which indicates that blood flow and the perfusion rate of the organ did not play a key role in the distribution of MHO7.

### 2.5. Changes of Gut Microbiome in Mice after Oral MHO7

To identify the effects in the compositional distribution of cecal microbiota induced by MHO7, the variable regions V3–V4 of 16S rRNA gene of the cecal samples from different time groups after oral MHO7 was sequenced by Illumina HiSeq/MiSeq platforms. In total, 760 million 400-bp paired-end reads were generated, with an average length of 437 bp of 1,262,525 sequences for each sample. To determine whether the sequencing adequately captured the diversity of the gut microbiota, rarefaction and Shannon index analysis was performed indicating that the sequencing depth was sufficient to cover most of the diversity (Appendix A). The results are presented as operational taxonomic units (OTUs) based on a sequence similarity of greater than 97% (Figure 4).

Principal coordinates analysis (PCoA) and Hierarchical clustering analysis revealed that a distinct clustering of the microbiota composition for each group (Figure 4A,B). After oral administration of MHO7, the microbial communities in the caeca of the MHO7-1 h group were more closely related to those of the control group, and the MHO7-8 h and MHO7-30 h groups had a distinct microbiota composition that clustered differently from the other two groups. At the OTU level, compared to the control group, the gut community diversity measured by the Shannon index decreased significantly in the MHO7-8 h group (Figure 4C; *p* < 0.05). A remarkable reduction of gut community richness measured by the Sobs index (Figure 4D; *p* < 0.01) and the amount of OTUs (Figure 4E; *p* < 0.001) appeared in MHO7-8 h and MHO7-30 h groups, compared to the control group. However, there was no significant difference in richness, diversity, or OUT number of the gut microbiome between control and MHO7-1 h group, indicating that the effect of oral MHO7 to gut microbiome of mice was influenced over time.

To analyze the differences among control and treatment groups, a supervised comparison of the microbiota between control and treatment groups was performed by linear discriminant analysis (LDA) and effect size analysis (LEfSe) without any adjustments, which are often used to identify the specific bacterial taxa among different groups. The greatest differences in taxa between each treatment time and the identified key phylotypes as microbiological markers at various phylogenetic levels are shown in Figure 5. The taxonomic levels of the circles from inside to outside are phylum, class, order, family, and genus. We used a logarithmic LDA score threshold > 2 to identify significant taxonomic differences between the control and drug groups (Appendix A). Since many bacteria in mice gut have not yet been identified at the species and even OTU levels, we analyzed gut microbiota from phylum to genus level.

Taxonomic representations of the significantly different taxa (cladogram) between control and treatment groups are shown in Figure 5, highlighting the relationship between taxa at different taxonomic levels in a tree-like structure and revealing how the significantly different taxa are interrelated. LEfSe revealed that the phylum and its derivative. After oral administration of MHO7 in treatment groups, the major changes were: Tenericutes, Saccharibacteria phylum, Bacteroidales_S24-7_group family and their immediate subordinate taxa at 1 h; Proteobacteria phylum, Bacilli class and its subordinate taxa at 8 h; and Deferribacteres, Verrucomicrobia phylum, Erysipelotrichia, and their subordinate taxa at 30 h. However, no significant differences were demonstrated for predominant taxa Firmicutes and Bacteroidetes after oral administration of MHO7.

To profile specific changes after oral MHO7 at different times in gut microbiota community of mice from a taxonomic perspective, the taxon tree from phylum to genus is constructed in Figure 6 based on the significant difference genera of the gut microbiota in mice by analyzing the relative abundance differences of four groups at different taxonomic levels in Appendix A and Appendix A.

At the phylum level, Firmicutes and Bacteroidetes were the dominant bacterial communities in all samples (Appendix A) and there were no significant differences after oral administration of MHO7 at 1, 8 or 30 h (Figure 6). However, other low abundance phyla, Proteobacteria, and Verrucomicrobia presented increased abundance in MHO7-8 h and -30 h groups in contrast to Tenericutes (Figure 6). Previous studies revealed that the increased abundance of Proteobacteria was associated with intestinal dysbiosis, metabolic disorders, inflammatory bowel disease, and lung diseases [20,21]. Verrucomicrobia was considered to be potential to induce regulatory immunity [22].

At the class level, MHO7 markedly diminished the level of Clostridia at 1 h (38.58 ± 11.1%) and 8 h (28.98 ± 20.11%) after oral administration when compared to control group (51.23 ± 18.09%), but the reduction was recovered at 30 h (47.82 ± 20.66%) (Figure 5 and Appendix A); in contrast, the relative abundance of Bacilli remarkedly enriched at 1 h and 8 h, and reversed at 30 h (Figure 5), which revealed that the influence of MHO7 on these classes was recoverable over time. The relative abundance of Mollicutes was significantly raised at 1 h and reduced at 8 and 30 h. Furthermore, when compared with control and MHO7-1 h groups, respectively, the increase of Verrucomicrobiae, Betaproteobacteria was noteworthy differences in MHO7-8 h and -30 h groups. At the order level, Clostridiales, Bacillales, Lactobacillales, Burkholderiales, Verrucomicrobiales, Mollicutes_RF9, and Anaeroplasmatales exhibited consistent trends at the class level.

At the family level, the relative abundance of Ruminococcaceae, Christensenellaceae, and Rikenellaceae displayed a sustained reduction at 1, 8 and 30 h after oral administration of MHO7 comparing with control group. The level of Lachnospiraceae and Family_XIII was declined in MHO7-1 h and -8 h groups but raised again in MHO7-30 h group when compared with control group. To the contrary, the relative abundance of Lactobacillaceae, Streptococcaceae, and Bacteroidales_S24-7_group was increased in MHO7-1 h and -8 h groups and decreased in MHO7-30 h group. The level of Peptococcaceae, Prevotellaceae, Alcaligenaceae, and Verrucomicrobiaceae was reduced at 1 h and increased at 8 and 30 h after oral administration of MHO7. On the contrary, norank_o__Mollicutes_RF9 and Anaeroplasmataceae were increased in MHO7-1 h group and decreased in MHO7-8 h and -30 h groups.

The relative abundance of Ruminococcaceae was enriched in breast cancer case patients, while the Lachnospiraceae were relatively less abundant [23,24]. The influence of MHO7 on downregulation of Ruminococcaceae and upregulation of Lachnospiraceae indicated that it has potential in the treatment of breast cancer. However, the families Lachnospiraceae, Ruminococcaceae, Christensenellaceae, Bacteroidaceae, Lactobacillaceae, and Rikenellaceae were reported to be associated with visceral fat, which resulted in cardio-metabolic disease risk and obesity [25,26,27]. Moreover, Mokkala et al. (2017) investigated that the relative abundance of the Ruminococcaceae might lead to a higher odds of positive gestational diabetes (GDM) in diagnosis [28], which revealed that MHO7 reducing the level of family Ruminococcaceae was beneficial to GDM. However, it is worth noting that the decrease of Ruminococcaceae and increase of Streptococcaceae might be a risk for cirrhosis [29] and IBD [30].

At the genus level, Figure 5 displays that 28 genera showed significant variation after oral administration of MHO7 with the dominated genera being *g__norank_f__Bacteroidales_S24-7_group*, *g__norank_f__Lachnospiraceae*, *Lactobacillus,* and *Rikenellaceae_RC9_gut_group* in all groups (Appendix A). The levels of *Coprococcus_1*, *g__norank_f__Ruminococcaceae*, *Ruminococcus_1*, *Ruminiclostridium_9*, *Ruminococcaceae_UCG-010*, *Ruminococcaceae_UCG-009*, *g__norank_f__Christensenellaceae*, *Papillibacter*, *Peptococcus, Ruminiclostridium,* and *Rikenellaceae_RC9_gut_group* were characterized by a continuous decline at 1, 8, and 30 h after oral administration of MHO7. The genera *g__norank_f__Lachnospiraceae*, *Lachnoclostridium*, *Sutterella,* and *[Eubacterium]_coprostanoligenes_group* decreased in MHO7-1 h group and enriched in MHO7-8 h and -30 h groups as compared with control group. In contrast, the genera *[Eubacterium]_ventriosum_group*, *Ruminococcaceae_UCG-014*, *Ruminococcaceae_NK4A214_group*, *norank_o__Mollicutes_RF,9* and *Anaeroplasma* increased at 1 h and reduced at 8 and 30 h after oral administration. The investigations indicated that, in high-fat diet-fed mice gut, the genera abundances of *Ruminiclostridium* and *Ruminococcaceae UCG-009* were elevated while *Ruminococcus_1, Ruminococcaceae_NK4A214_group*, *Ruminococcaceae_UCG-009*, *Ruminococcaceae_UCG-010,* and *Ruminiclostridium_9* were decreased [31,32,33]. However, in the isoproterenol-induced acute myocardial ischemia model, the amounts of *Rikenellaceae RC9 gut group*, *Ruminococcus 1*, and *Bacteroidales S24-7 group* were elevated while *Ruminiclostridium 9*, *Lachnoclostridium,* and *Ruminococcaceae UCG-014* were reduced. The enrichments of *Ruminiclostridium 5* and *Rikenellaceae RC9 gut group* were considered to be related to the lipid metabolism [34,35]. Furthermore, genus *Ruminiclostridium 9* was positive to blood IgM level and colitis histological scores [36] and *Anaeroplasma* was supposed to be a potential anti-inflammatory probiotic for the treatment of chronic intestinal inflammation [37]. The researchers illustrated that, although the impact of MHO7 on these genera was not entirely consistent as reported but it mediated intake metabolism and inflammatory responses.

In addition, the amounts of *g__norank_f__Bacteroidales_S24-7_group*, *Prevotellaceae_UCG-001*, *Lactobacillus*, *Streptococcus,* and *Prevotellaceae_NK3B31_group* was enriched in MHO7-1 h and -8 h groups and reduced in MHO7-30 h group compared with control group. The reduction of genus *Lactobacillus* was associated with cardiovascular diseases (CD), inflammatory bowel disease (IBD), and chronic kidney disease [29,38], illustrating that MHO7 was beneficial to these diseases. Generally, most of the intestinal microbes influenced by MHO7 were the producers of short-chain fatty acids (SCFA), mainly acetate, propionate, and butyrate, which are believed to play a beneficial role in human gut health [39]. The correlation analysis illustrated that the genera *Ruminococcaceae_UCG-010* and *Ruminococcaceae_UCG-014* had a negative correlation with SCFA [31,40] while the *Ruminococcaceae NK4A214 group* and *Ruminococcaceae UCG-005* had a positive correlation with the SCFA [33,41]. Bacteroidales_S24-7_group family produces acetate, propionate, and succinate [42] and *Coprococcus_1* genus can reduce the production of propionic acid [43]. In conclusion, viewed from intestinal microflora, MHO7 influenced the metabolic pathways by SCFA through regulating the level of gut microbiome.

## 3. Materials and Methods

### 3.1. Chemicals and Reagents

MHO7 (purity ≥ 98%) was obtained from the fermentation products of *Aspergillus*
*ustus* isolated from mangrove rhizosphere. The compound was isolated by column chromatography and the purity was controlled by HPLC using the standard curves for quantitative detection. Progesterone (IS), acetonitrile and methanol (HPLC grade), n-octanol, pepsin, trypsin, sodium dihydrogen phosphate, disodium hydrogen phosphate, dilute hydrochloric acid, sodium hydroxide, potassium dihydrogenphosphate, and dipotassium hydrogenphosphate were purchased from Sigma-Aldrich (St. Louis, MO, USA). Ultrapure water was obtained from distilled water using a Milli-Q system (Millipore, Milford, MA, USA).

### 3.2. Instrumental Analysis Method of HPLC and LC-MS

#### 3.2.1. HPLC Conditions

An HPLC system (Shimadzu, Japan) was equipped with a LC-20AD binary pump system, a SIL-20AD auto sampler, a CBM-20A system controller, and a CTO-20A oven. Separation of MHO7 and IS from endogenous substances was carried out on an Agilent Zorbax XDB C18 column (5 μm, 250 × 4.6 mm, Agilent, Pal Alto, CA, USA), which was kept at 30 °C. The mobile phase was composed of 30% phase A (water) and 70% phase B (acetonitrile) with a flow rate of 1 mL·min^−1^. The inject volume of the sample solution was 10 μL.

#### 3.2.2. LC-MS/MS Conditions

Chromatographic separation was achieved using the HPLC system (Thermo Scientific Accela™, Thermo Fisher Scientific, San Jose, CA, USA), equipped with a quaternary pump system, a de-gasser, a diode-array detector, an autosampler, and a column oven. Separation of MHO7 and IS from endogenous substances was carried out on a Waters CORTECS C18 column (2.7 μm, 150 × 4.6 mm, Waters, Milford, MA, USA), which was kept at 30 °C. The mobile phase is composed of phase A (water) and phase B (acetonitrile) with a flow rate of 0.4 mL·min^−1^. The gradient elution used to achieve chromatographic separation was as follows: 0–5 min, 10% B; 5–8 min, 10–70% B; 8–12 min, 70% B; 12–18 min, 70–95% B; 18–27 min, 95% B; and 27–30 min, 95–10% B. The inject volume of the sample solution was 10 μL and the autosampler temperature was maintained at 10 °C.

Mass spectrometric analysis was carried out on a Thermo Scientific TSQ Quantum MS/MS system equipped with an electrospray ionization interface (ESI). Compound dependent parameters and instrumental parameters were optimized by infusing neat solutions of MHO7 and the IS separately by using a syringe pump. A selective-reaction monitoring (SRM) mode was applied for the detection of the transition of m/z 367.3 → 349.2, 307.2 m/z for MHO7 and m/z 315.4 → 109.2, 97.0 m/z for progesterone (IS), respectively. The typical operating source conditions for MS scan in positive ion ESI mode were optimized as follows: sheath gas flow rate, 20; Aux gas flow rate, 5; spray voltage, 3.50 kV; capillary temperature, 320 °C; tube lens, 75.2 V of MHO7 and 89.1 V of IS; and collision energy, 13 V and 11 V of MHO7 (product ion 349.2, 307.2 m/z) and 28 V and 20 V of IS (product ion 109.2, 97.0 m/z), respectively.

### 3.3. Bioanalytical Method Validation

The LC–MS/MS method was validated according to the U.S. Food and Drug Administration (FDA) guidance for bioanalytical method validation [44].

#### 3.3.1. Specificity and Selectivity

The specificity and selectivity of the method were investigated by the screening analysis of incubation system solutions and six individual blank mice plasma and tissues samples. Two other incubation system solutions and blank plasma and tissues samples containing an IS concentration of 5.0 mM·L^−1^ were also determined for interference. Each blank sample was tested for exclusion of endogenous interference at the retention times of MHO7 and IS.

#### 3.3.2. Linearity and Lower Limit of Quantification (LLOQ)

The linearity of the method was evaluated by using six calibration standards over a calibration range of 0.01–2 mM·L^−1^ in oil–water system, 0.01–5 μM L^−1^ in the in vitro incubation system (SGF), 0.01–5 μM·L^−1^ in plasma, 0.005–5 and 0.05–50 μM·L^−1^ in gastrointestinal contents, and 0.005–5 and 0.5–50 μM·L^−1^ in mice liver. LLOQ was determined as the lowest concentration of analytes with %CV (coefficient of variation) not exceeding 20% and accuracy in the range of 80–120%.

#### 3.3.3. Precision and Accuracy

Intra- and inter-batch precision and accuracy of the developed method were investigated by analyzing QC samples at four different concentrations for six replicates. The precision of the method was determined by %CV and accuracy was evaluated by recovery. As per US-FDA guidelines the acceptable limit of %CV was < 20% for LLOQ and ≤ 15% for HQC, MQC, and LQC. The accuracy was calculated as percent difference in mean value of the observed concentration and nominal concentrations of QC samples.

#### 3.3.4. Extraction Recovery and Matrix Effect

The extraction recovery and matrix effects were investigated at HQC, MQC, and LQC concentration levels. Extraction recovery of MHO7 in SGF, mice plasma, and tissues was evaluated by comparing peak area ratios of MHO7 to IS of an extracted sample (*n* = 6) to the standard analytes solution of same concentration. To study matrix effect, initially, blank SGF, plasma, and tissue samples were processed followed by spiking of the analyte MHO7 and IS to the post processed samples. Further, aqueous solutions of analyte and IS of the same concentrations were prepared and analyzed. The matrix effect was calculated using the equation in the range of 0.8–1.2.

#### 3.3.5. Stability Studies

To evaluate the stability of MHO7 in SGF, plasma, and tissues samples, six replicates of high- and low-quality control samples were analyzed. Post preparative stability was carried out at 10 °C for 48 h. The bench top and long-term stability were analyzed at ambient temperatures 25 °C for 48 h, and −80 °C for 30 days, respectively. For freeze–thaw stability, quality control samples were stored at −80 °C. After three freeze–thaw cycles, quality control samples were analyzed. The stability was assessed by comparing with nominal concentration of the analyte at 0 h and the mean percentage changes are within the acceptance criteria of ± 10%.

### 3.4. Determination of Log P and Log D Value of MHO7 by Shaking Flask Method

Log P value of MHO7 between n-octanol and water was determined at 25 and 37 °C, respectively. Log D value of MHO7 between n-octanol and phosphate buffer under three different pH (pH 1.5, 5.0, and 7.4) systems were determined at 37 °C. The phases of distilled water and n-octanol were saturated with each other by shaking for 48 h with 250 rpm in a shaking water bath prior to use. Solutions of MHO7 was prepared in concentration of 10 mM·L^−1^ in n-octanol. Accurately measured amounts of the two solvents with necessary quantity of MHO7 were placed in a 5.0 mL centrifuge tube; then, the mixture was carried out at 250 rpm and designated temperature in the shaking water bath for 48 h to afford complete phase separation. The samples of both phases were collected at 0, 4, 8, 12, 24, 30, 36 and 48 h, diluted with methanol (1:1, V) and determined by HPLC with the detection wavelength of 234 nm. Each set of solvent systems for experimental logP/D value was the mean of the three determinations.

### 3.5. SGF and SIF Stability of MHO7

SGF was prepared with modifications according to Lee et al. (2012) by mixing 100 mL of distilled water with a pH value set to 2.0 with hydrochloric acid and 1 g pepsin solution (800–2000 U/mg of protein). SIF was produced according to USP specifications (Test Solutions, United States Pharmacopeia 35, NF 30, 2012). Monobasic potassium phosphate (0.68 g) was dissolved in 25 mL of water. Then, 7.7 mL of 0.2 M NaOH was added to adjust the pH to 6.8. To this, 1 g of pancreatin was added and shaken gently until dissolved and the volume adjusted to 100 mL with water. Pancreatin was added after adjusting the pH of the solution to 6.8 to avoid precipitation of the enzyme.

Aliquots (199 µL) of SGF and SIF were placed in 1.5 mL microcentrifuge tubes and incubated at 37 °C for 10 min in a water bath, respectively. Different concentrations of MHO7 (1 µL) were added to each of the above microcentrifuge tubes to start the reaction with being incubated at 37 °C at 250 rpm in a shaking water bath. The reaction was stopped by adding ice ACN (800 µL) to each tube at intervals of 0, 0.5, 1, 2, 4, 6, and 8 h. The samples were vortexed for 15 min and centrifuged at 4 °C and 15,000 rpm for 15 min. After centrifuging the supernatant layer was taken directly for LC-MS analysis. Each experiment was carried out in triplicate, and average values were plotted. QC samples were prepared in the same way of inactivating pepsin in SGF incubation system at the concentrations of 1, 0.1 and 0.01 μM·L^−1^ for HQC, MQC and LQC, respectively.

### 3.6. Animals and Treatment

Male and female KM mice of 8–10 weeks of age (20–22 g) were used in acute toxicity experiment and only female KM mice were subjected in other animal experiments. All mice were purchased from the Vital River Laboratory Animal Technology Co. Ltd. (Beijing, China). The mice were bred in a standard environment (23 ± 1 °C; humidity of 60 ± 5%) with a 12-h light/dark cycle and food and water were provided ad libitum. All animal protocols were approved by the Animal Care and Use Committee of Wuhan University Center for Animal Experiment ABSL-Ⅲ Laboratory and complied with the Guide for the Care and Use of Laboratory Animals (NIH publication, 8th edition, 2011). In total, 130 mice were used for in vivo studies under the ethical approved protocols.

### 3.7. Determination of the Maximum Tolerated Dose (MTD) of MHO7 in Mice

Male and female KM mice of 8–10 weeks of age (20–22 g) were used in acute toxicity experiment. After animals were acclimatized for 1 week, 20 female and 20 male mice were randomly and evenly divided into four groups. The mice of acute toxicity experiment were fasted overnight and administered by the oral dose of MHO7 (2400 mg·kg^−1^, bw) diluted in corn oil the next morning. Then, mice were placed with free access to chow and water. The mice were observed for two weeks in all groups, and the weight was recorded at 0, 1, 7, and 14 days. The number surviving and their behaviors was also recorded every day.

### 3.8. Pharmacokinetic Studies Assay

Ninety female KM mice of 8–10 weeks of age (20–22 g) were divided randomly into 15 groups, and, after fasting for 12 h, the mice in other experiments were oral administrated at the dose of 500 mg·kg^−1^ bw. About 500 μL blood samples were collected at 0.5, 1, 2, 3, 4, 5, 6, 8, 10, 12, 16, 20, 24, 30 and 36 h into heparinized tubes after the intragastric administration of MHO7. All blood samples were immediately centrifuged at 8000 *g* for 5 min and the resulting plasma was transferred and then processed as described in sample preparation for the analysis by LC-MS/MS. MHO7 levels in plasma were quantified by HPLC–MS/MS. The calibration curve was constructed by plotting the peak area ratio of MHO7 to the internal standard and the linearity was determined by weighted (1/x) linear regression analysis. The regression equation of the calibration curve was then used to calculate the concentration of MHO7 in the plasma. QC samples and standard curve series solutions were prepared in the same way of blank plasma. The concentrations of HQC, MQC, and LQC were 5, 0.5, and 0.05 μM·L^−1^, respectively, and the concentrations of standard curve series solutions were 5, 2.5, 1, 0.5, 0.25, 0.1, 0.05, and 0.01 μM·L^−1^.

### 3.9. Tissue Distribution Study Assay

The animals and treatment methods were the same as in the pharmacokinetic studies. Afterwards, the biological samples (*n* = 6 per group) were collected at 1, 4, 8, 12, 20, and 30 h, and the tissues including heart, liver, spleen, lung, kidney, stomach, intestine, brain, muscle (hind limb), fat (white adipose tissue ), reproductive organs (uterus, fallopian tube and ovary) stomach and its contents, and intestine and its contents, and the blood samples were simultaneously collected. All the tissues were washed with cold physiological saline (4 °C), excess fluid blotted, accurately weighed, and subjected to the processing as described in the sample preparation. QC samples and standard curve series solutions were prepared in the same way of blank gastrointestinal contents and livers, respectively. The concentrations of HQC, MQC, and LQC in mice tissues were 10, 1, and 0.1 μM·L^−1^, respectively, and the concentrations of standard curve series solutions were 50, 25, 10, 5, 2.5, 1, 0.5, 0.25, 0.1, and 0.005 μM·L^−1^ for high and low concentrations of standard curves.

### 3.10. Gut Microbiota Analysis

#### 3.10.1. Sample Collection

Fresh cecal samples from 24 (*n* = 6 per group) mice were collected at 1, 8 and 30 h after oral administration of 500 mg·kg^−1^ MHO7 meanwhile control group was collected at 8 h after oral the same volume corn oil, respectively. All of the cecal samples were collected in sterile environments and containers, snap-frozen in liquid nitrogen, and then stored at −80 °C.

#### 3.10.2. DNA Extraction, PCR Amplification and Sequencing

Microbial DNA was extracted from 24 cecal samples using the E.Z.N.A.® soil DNA Kit (Omega Bio-tek, Norcross, GA, USA) according to manufacturer’s protocols. The final DNA concentration and purification were determined by NanoDrop 2000 UV-vis spectrophotometer (Thermo Scientific, Wilmington, DE, USA), and DNA quality was checked by 1% agarose gel electrophoresis. The V3-V4 hypervariable regions of the bacteria 16S rRNA gene were amplified with primers 338F (5’- ACTCCTACGGGAGGCAGCAG-3’) and 806R (5’-GGACTACHVGGGTWTCTAAT-3’) by thermocycler PCR system (GeneAmp 9700, ABI, Foster City, CA, USA). The PCR reactions were conducted using the following program: 3 min of denaturation at 95 °C; 27 cycles of 30 s at 95 °C, 30 s for annealing at 55 °C, and 45 s for elongation at 72 °C; and a final extension at 72 °C for 10 min. PCR reactions were performed in triplicate 20 μL mixture containing 4 μL of 5 × FastPfu Buffer, 2 μL of 2.5 mM·L^−1^ dNTPs, 0.8 μL of each primer (5 μM·L^−1^), 0.4 μL of FastPfu Polymerase, and 10 ng of template DNA. The resulted PCR products were extracted from a 2% agarose gel and further purified using the AxyPrep DNA Gel Extraction Kit (Axygen Biosciences, Union City, CA, USA) and quantified using QuantiFluor ™ -ST (Promega, Madison, WA, USA) according to the manufacturer’s protocol.

Purified amplicons were pooled in equimolar and paired-end sequenced (2 × 300) on an Illumina MiSeq platform (Illumina, San Diego, CA, USA) according to the standard protocols by Majorbio Bio-Pharm Technology Co. Ltd. (Shanghai, China).

#### 3.10.3. Bioinformatics Analysis

Raw fastq files were quality-filtered by Trimmomatic and merged by FLASH according to the following criteria: (i) The reads were truncated at any site receiving an average quality score < 20 over a 50 bp sliding window. (ii) Sequences whose overlap was longer than 10 bp were merged according to their overlap with mismatch no more than 2 bp. (iii) Sequences of each sample were separated according to barcodes (exactly matching) and Primers (allowing 2 nucleotide mismatching), and reads containing ambiguous bases were removed. Operational taxonomic units (OTUs) were clustered using UPARSE (version 7.1, http://drive5.com/uparse/) at a 97% similarity level with a novel “greedy” algorithm that performs chimera filtering and OTU clustering simultaneously. The taxonomy of each 16S rRNA gene sequence was analyzed by RDP Classifier algorithm (http://rdp.cme.msu.edu/) against the Silva (SSU123) 16S rRNA database using confidence threshold of 70%.

### 3.11. Statistical Analysis

Data for each group are presented as mean ± SD. Statistical significance of different formulations were measured by one-way analysis of variance (ANOVA) followed by the Tukey Kramer multiple comparison test, using SPSS software version 14.0 (SPSS, Chicago, IL, USA), and the statistical significance was expressed by a p-value of less than 0.05. The main pharmacokinetic parameters of MHO7 was fitted by the average mean of a group in each time point with the Phoenix WinNonLin software version 6.3 (Pharsight, Cary, NC, USA). The gut microbiome analysis was performed using the free online platform of Majorbio Cloud Platform (www.majorbio.com).

## 4. Conclusions

To the best of our knowledge, this is the first report to evaluate the toxicity, pharmacokinetics, and gut microbiome of sesterterpene MHO7. The oil–water distribution coefficients was characterized by Log P/D values of MHO7 ranging 0.93–2.48, which is satisfactory for oral preparation [19].

The maximum tolerable dose (MTD) of MHO7 was 2400 mg·kg^−1^ in mice by oral administration. A rapid, reliable, and sensitive HPLC–MS/MS method was validated for quantitative analysis of MHO7 in mouse biological samples such as plasma and tissues. MHO7 was rapidly distributed in mouse plasma and tissues following a single oral administration. The major target tissue depots of MHO7 in mice was the reproductive organs including uterus, ovary, and oviduct, and the renal excretion might be the main elimination route for MHO7. Additionally, the high concentration of MHO7 in fat is worthy of attention and there was no long-term accumulation in the tissues.

After oral administration of MHO7 for 8–30 h, the diversity and richness of gut community were significantly reduced. Although MHO7 had little influence on the level of dominant microbes at the phylum level, we observed that the relative abundance of 28 genera showed significant change among control and treatment groups. Most of the genera influenced belong to Firmicutes (21 of 28), most of these genera (18 of 21) under the order Clostridiales, class Clostridia. The remarkably changed genera classified to clostridia came from family Ruminococcaceae (11 of 18) and Lachnospiraceae (4 of 18). The downregulation of Ruminococcaceae and upregulation of Lachnospiraceae at family level of MHO7 has potential in the treatment of breast cancer [23,24], but the negative effects on the intestine should be noted.

## Figures and Tables

**Figure 1 marinedrugs-17-00667-f001:**
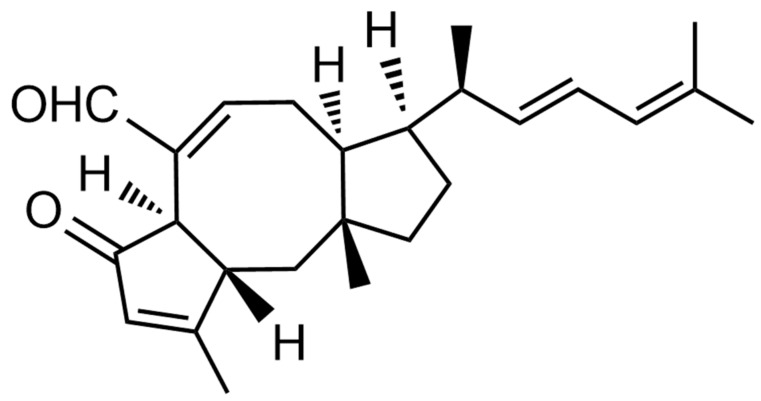
The structure of MHO7.

**Figure 2 marinedrugs-17-00667-f002:**
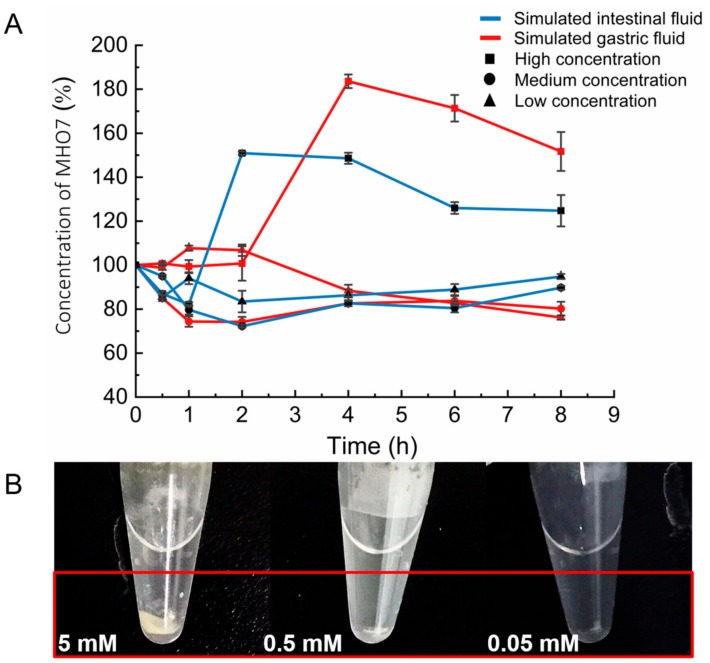
Stability of MHO7 in Simulated Gastric Fluid (SGF) and Simulated Intestinal Fluid (SIF): (**A**) effects on incubation system of different concentrations of MHO7; and (**B**) stability of different concentrations of MHO7 in SGF and SIF (*n* = 3 per group).

**Figure 3 marinedrugs-17-00667-f003:**
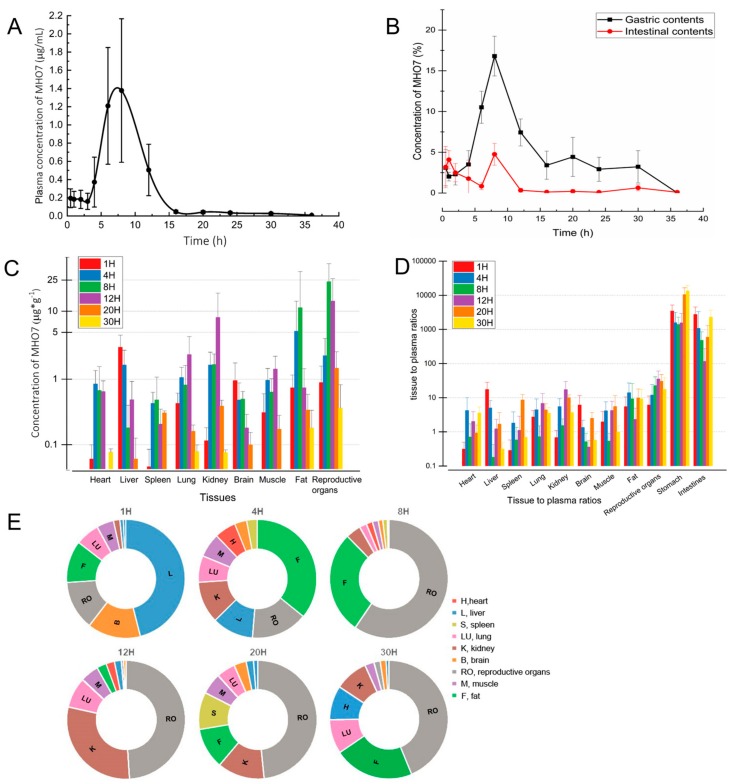
Pharmacokinetics and tissue distribution study of MHO7 in female mice at a signal oral administration dose of 500 mg·kg^−1^ (*n* = 6 per group): (**A**) the plasma concentration–time curve of MHO7; (**B**) the percentage concentrations of MHO7 in the stomach and intestine at different time points; (**C**) the concentrations of MHO7 in tissues; (**D**) the ratios of tissues to plasma of MHO7; and (**E**) the pie chart of concentrations of MHO7 in tissues at different time points.

**Figure 4 marinedrugs-17-00667-f004:**
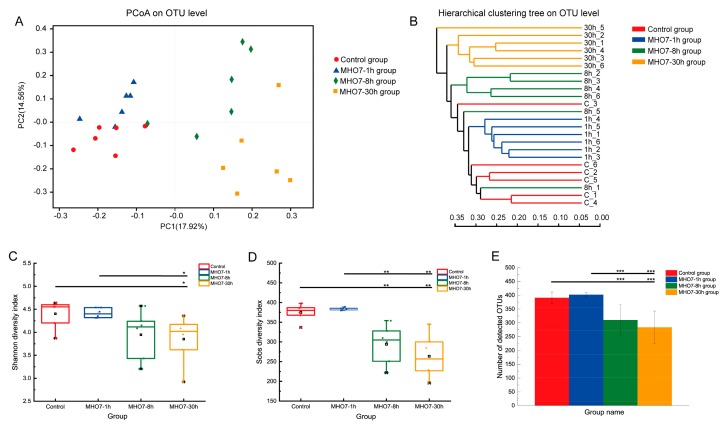
Modulation of structure and diversity of gut microbiota in different times after oral administration of MHO7 (*n* = 6 per group): (**A**) principal coordinate analysis (PCoA) and (**B**) sample clustering results of the unweighted UniFrac distances of microbial 16S rRNA sequences from the V3–V4 region; (**C**) alpha diversity analysis at the OTU level of Shannon index; (**D**) alpha diversity analysis at the OTU level of SOBS index; and (**E**) number of detected OTUs. Significant differences between control and drug groups are indicated: * *p* < 0.05; ** *p* < 0.01; *** *p* < 0.001.

**Figure 5 marinedrugs-17-00667-f005:**
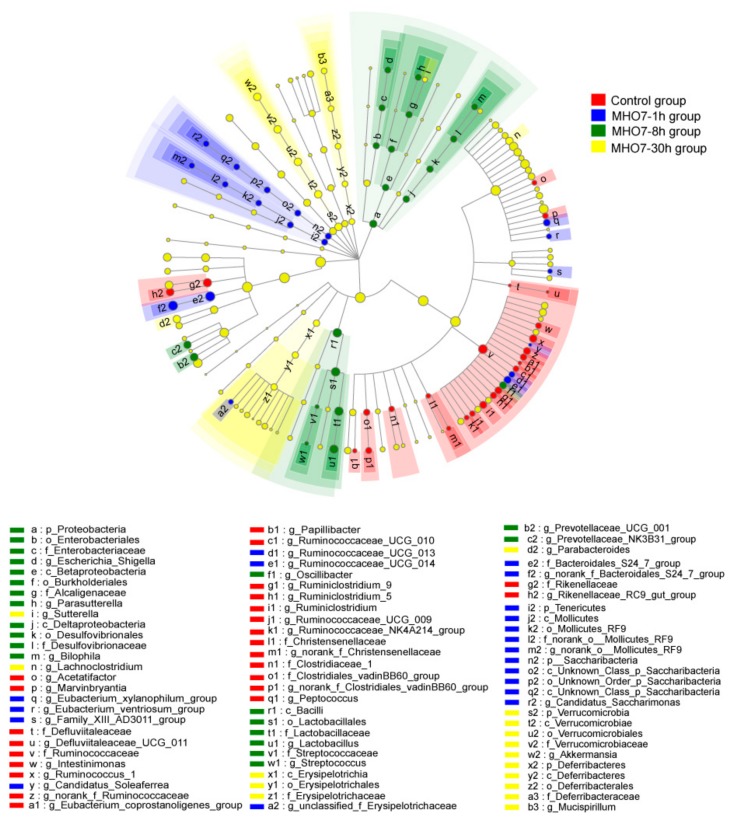
LEfSe results on gut microbiota in drug groups compared with control group. Difference are represented in the color of the most abundant class. Each circle’s diameter is proportional to the taxon’s abundance.

**Figure 6 marinedrugs-17-00667-f006:**
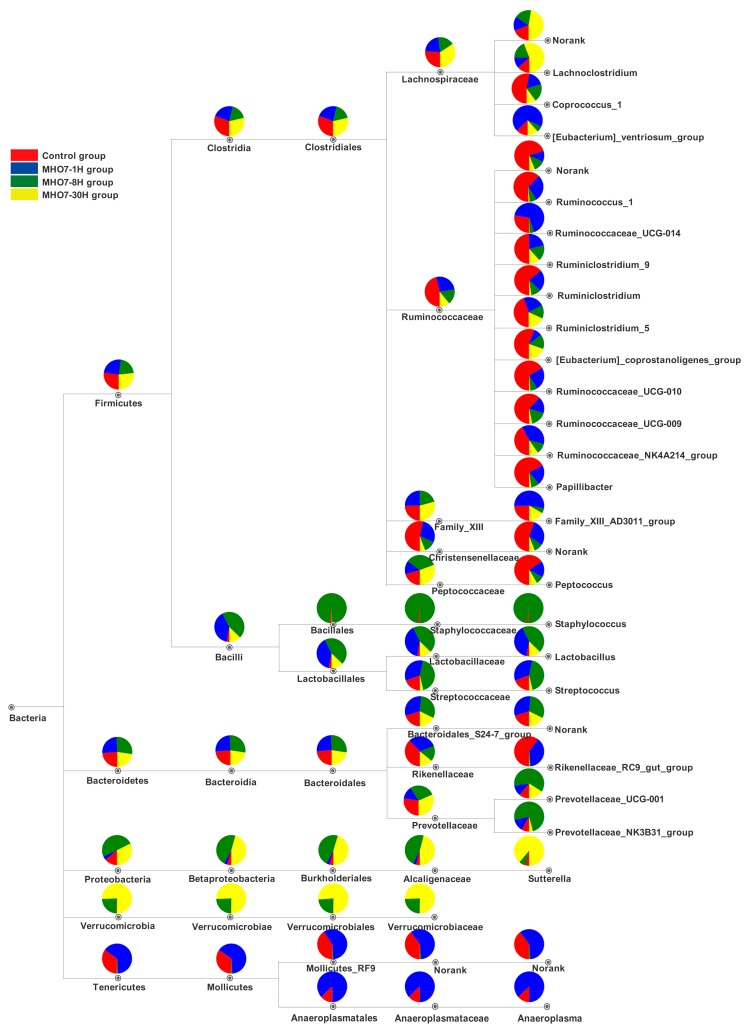
The taxon tree created based on the significantly different genera of the gut microbiota in female mice after oral administration of MHO7 at different time points of a signal dose of 500 mg·kg^−1^.

**Table 1 marinedrugs-17-00667-t001:** Main pharmacokinetic parameters of MHO7 in mice plasma.

Parameter	Estimate
Lambda_z	0.10·h^−1^
t_1/2_	6.97 h
T_max_	8.00 h
C_max_	1.38 μg·mL^−1^
AUC	10.50 h·μg·mL^−1^
V/F	479.02 L·kg^−1^
CL/F	47.61 L·h^−1^·kg^−1^
MRT	8.76 h

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
