# Peer review of "Toxicity, Pharmacokinetics, and Gut Microbiome of Oral Administration of Sesterterpene MHO7 Derived from a Marine Fungus"

_marinedrugs, 2019, doi:10.3390/md17120667_

Round 1
Reviewer 1 Report
The manuscript by Tian et al. reports an in vivo toxicity and pharmacokinetics studies as well as an effect on gut microbiome of the sesterterpene MHO7. The study was conducted using the in vivo mice model. Current research is well designed, the manuscript is well written and the data is well presented. However I completely miss the reason of this study. In the previous very recent paper of same group (Pharmacol. Res. 2019, 146, P. 104294 ) it has been reported an anticancer in vitro effect in breast cancer cells as well as the effect of the compound on ER and its signaling. Next logical step on the way to the clinically approved drug would be an evaluation of in vivo anticancer effect, following by the third step – evaluation of the side effects, toxicity, pharmacokinetics and things like an effect on gut microbiome. However, in the current story a second step is missing, which raises a reasonable question – why this research was conducted? Did the authors evaluate an antitumor activity in vivo? If so the authors should mention it here, even though the results may have been negative (which from my point of view should not be an obstacle for the publication of the current manuscript).
Additionally, how the side effects were evaluated? Has the blood cell count been performed? Has the weight of the organs been examined (in particular, liver and kidneys, as the renal excretion has been suggested as a main elimination route of the drug). Why the one-time drug administration regime was chosen – does it anyhow represent the administration regime which showed any therapeutic effect before?
Additional minor points:
- Please provide the number of the permission to provide the described above animal experiments (especially taking in account such a great number of mice used).
- Please provide the structure of the tested compound in the manuscript.
- Should the “water-like stool” be called diarrhea or was it something different?
- Materials and methods: how was the MHO7 isolated? How the purity was controlled?
Author Response
Dear reviewer:
Thank you very much for your important comments concerning our manuscript. We have studied the valuable comments from you including research route, accuracy of words and completeness of details and so on. We tried our best to revise and improve the manuscript and hoped to get your approval. Depending upon your advice, we amended the relevant part in manuscript. All revisions were marked in the manuscript by using the track changes mode in MS Word. The point to point responds to your comments are listed as following:
General Comment 1: Did we evaluate the second step about the antitumor activity in vivo?
Response: Thank you for your valuable advice. It is very necessary to evaluate pharmacodynamics in vivo before the third step of evaluation of the side effects, toxicity, pharmacokinetics. Actually, we constructed the breast cancer model in nude mouse with MCF-7 cell lines to evaluate the antitumor activity of MHO7 in vivo. The results indicated that the antitumor activity of MHO7 with the dose of 20 mg/kg was equivalent to paclitaxel with the dose of 10 mg/kg. The pharmacodynamic experiments are still in progress and the data was not show here.
General Comment 2: How the side effects were evaluated?
Response: Thank you for your valuable suggestion. We evaluated the side effects with the weight of the organs including heart, liver, spleen, lung, kidney, brain, reproductive organ (testicle / uterus) and the results suggested that there were no significant differences between treatment group and control group. Considering your suggestion, we revised the manuscript by adding the weight of the organs of result part (line 106-108 in modified files) and add the organ weight data s in supplementary materials (Table S5).
General Comment 3: Why the one-time drug administration regime was chosen?
Response: We agree that it is very important and necessary to study the multiple administration regimes under the therapeutic effect. However, in this study, we try to choose a dose which was approximate 1/5 of the MTD value of oral administration, and much higher than the anti-tumor dose in the pharmacokinetic study which is 20 mg/kg. We want to obtain more information about the compound distribution in organs and effects on gut microbiota. This is not common as usual, we hope you may agree.
Additional Comment 1: Please provide the number of the permission to provide the described above animal experiments.
Response: Thanks for your careful comment. 40 mice were used in the acute toxicity experiment and 90 mice were used to determine the plasma concentration time curve. The samples for tissue distribution and gut microbiota experiments were collected from these 90 mice. We use as few animals as possible to get more pharmacokinetic data. According to your suggestion, we also modified the expression of the number of experimental animals in the material method (line 475-477 in modify files), and uploaded the relevant license certificate in other independent supplementary material.
Additional Comment 2: Please provide the structure of the tested compound in the manuscript.
Response: Thanks for your suggestion. We have added the structure of MHO7 in the manuscript (line 52-53 and Fig. 1 in modify files).
Additional Comment 3: Should the “water-like stool” be called diarrhea or was it something different?
Response: Thanks for your comment of language expression. We corrected the word “water-like stool” to “diarrhea” in line 108 of modify files .
Additional Comment 4: Materials and methods: how was the MHO7 isolated? How the purity was controlled?
Response: Thanks for the valuable advice. The compound was isolated by column chromatography and the purity was controlled by HPLC using the standard curves for quantitative detection. We added the content to the materials and methods in line 341-342 of modify files.
We tried our best to improve the manuscript and made some changes in the manuscript. All revisions were marked in the manuscript by using the track changes mode in MS Word. We appreciate very much for your warm and careful work earnestly, and hope that the correction will meet with approval. Once again, thanks to you for your kind comments and suggestions.
Sincerely yours,
Wei Tian & Kui Hong
Reviewer 2 Report
The manuscript titled ‘Toxicity, Pharmacokinetics, and Gut microbiome of Oral Administration of Sesterterpene MHO7 Derived from a Marine Fungus’ is a well written manuscript. There are few minor comments that need to be addressed:
Figure 1: Statistics is missing Table 1: Standard deviation or standard error of means must be reported. Cmax of the compound looks low. Authors didn’t do any experiment or discuss the potential mechanism of low bioavailability. This needs to be improved.Author Response
Dear reviewer:
Thank you very much for your important comments concerning our manuscript. We have studied the valuable comments from you including discussion on experimental results and completeness of details in data and so on. We tried our best to revise and improve the manuscript and hoped to get your approval. Depending upon your advice, we amended the relevant part in manuscript. All revisions were marked in the manuscript by using the track changes mode in MS Word. The point to point responds to the your comments are listed as following:
Comment 1: Figure 1: Statistics is missing.
Response: Thank you for your careful advice. The statistical analysis for every time point was exhibited with the mean and standard deviation, maybe the error lines were not be obvious in the Fig.1. According to your suggestion, we adjusted the Fig.1 for more clear of the error lines (line 95-96 in modify files).
Comment 2: Standard deviation or standard error of means must be reported in Table 1.
Response: Thank you for your careful suggestion. In the pharmacokinetic experiments, plasma and tissue samples of a mouse were collected at the same time. The pharmacokinetic parameters were fitted by the mean value of a group, so the standard deviation or standard error of means was not showed in Table 1. We are sorry for the unclear statement in materials and methods department and modified it in statistical analysis (line 528-530 in modify files).
Comment 3: Cmax of the compound looks low. Authors didn’t do any experiment or discuss the potential mechanism of low bioavailability. This needs to be improved.
Response: Thank you for your valuable advice. We actually need discuss the potential mechanism of low bioavailability due to the low Cmax of MHO7. Considering your suggestion, we discussed the potential mechanism of low bioavailability of MHO7 in line and hoped to get your approval (line 136-141 in modify files).
We tried our best to improve the manuscript and made some changes in the manuscript. All revisions were marked in the manuscript by using the track changes mode in MS Word. We appreciate very much for your warm work earnestly, and hope that the correction will meet with approval. Once again, thanks to you for your kind comments and suggestions.
Sincerely yours,
Wei Tian & Kui Hong